# Assessment of Pharmaceutical Services for Smoking Cessation: An Effectiveness–Implementation Hybrid Study

**DOI:** 10.3390/ijerph191912305

**Published:** 2022-09-28

**Authors:** Maria Eduarda Pinheiro Laborne-e-Valle, Ana Emília de Oliveira Ahouagi, Debora Gontijo Braga, Isabela Vaz Leite Pinto, Célio Rezende Lara-Júnior, Sabrina Gonçalves Ferreira, Paula de Fátima Fernandes Blunk, Adriano Max Moreira Reis, Edna Afonso Reis, Djenane Ramalho-de-Oliveira, Mariana Martins Gonzaga do Nascimento

**Affiliations:** 1Center for Pharmaceutical Care Studies, College of Pharmacy, Federal University of Minas Gerais, Belo Horizonte 31270-901, Minas Gerais, Brazil; 2Belo Horizonte Municipality, Belo Horizonte 30130-003, Minas Gerais, Brazil; 3Pharmaceutical Products Department, College of Pharmacy, Federal University of Minas Gerais, Belo Horizonte 31270-901, Minas Gerais, Brazil; 4Statistics Department, Exact Sciences Institute, Federal University of Minas Gerais, Belo Horizonte 31270-901, Minas Gerais, Brazil

**Keywords:** smoking, smoking cessation, pharmaceutical services, primary health care, implementation science

## Abstract

Smoking is the main preventable cause of illness and early death worldwide. Thus, it is better to promote smoking cessation than to treat tobacco-related diseases. The objective of this study was to assess the implementation and effectiveness of smoking cessation pharmaceutical services offered in primary health care (PHC) in a large Brazilian city through a type 1 effectiveness–implementation hybrid study. The services were offered through individual or group approaches (Jan/2018–Dec/2019). The service indicators were described and the incidence of cessation in the services was evaluated. Factors associated with cessation were assessed by Poisson regression analysis. The services were offered in most PHC centers (61.2%) and by most pharmacists (81.3%). In total, 170 individual (9.7%) and 1591 group (90.3%) approaches occurred, leading to cessation in 39.4% (*n* = 67) and 44.8% (*n* = 712) of these, respectively. The use of nicotine plus antidepressants (RR = 1.30; 95%CI = 1.08–1.57; *p* = 0.006) and the number of sessions with pharmacists (RR = 1.21; 95%CI = 1.19–1.23; *p* < 0.001) were positively associated with cessation; a very high level of dependence was negatively associated (RR = 0.77; 95%CI = 0.67–0.89; *p* = 0.001). The smoking cessation services were effective and should be encouraged.

## 1. Introduction

Smoking is recognized as an epidemic chronic neurobehavioral disease, caused by physical, psychological, and behavioral dependence on nicotine [1,2]. It is part of the group of mental and behavioral disorders due to the use of psychoactive substances, constituting a risk factor for the development of various types of cancer and other illnesses [3,4].

It is estimated that, in the world, more than 7 million deaths result from direct tobacco use and approximately 1.2 million are the result of non-users of tobacco exposed to secondhand smoke [1,5]. Thus, smoking is considered the main preventable cause of illness and early death in the world and secondhand smoke would be the third leading cause of mortality [1]. The prevalence of smoking is decreasing in many countries. However, given its relevance as a public health problem, its prevalence still represents high numbers worldwide, especially in low- and middle-income countries such as Brazil [4,5,6].

Considering that it is less costly to help tobacco users to quit than to treat the tobacco-related diseases, smoking cessation is one of the most relevant health interventions [7]. Studies have shown the effectiveness of pharmaceutical services for smoking cessation [8,9,10]. However, to our knowledge, there is still a lack of studies demonstrating the impact of the pharmacist’s role in smoking cessation in primary health care (PHC), and it is still necessary to expand such knowledge, especially with “real-world” studies.

Thus, this study aimed to assess the implementation and effectiveness of pharmaceutical services for smoking cessation offered in the public PHC of a large Brazilian capital as well as the associated factors with smoking cessation.

## 2. Materials and Methods

### 2.1. Study Design

This is a type 1 effectiveness–implementation hybrid study according to the methods encouraged by the World Health Organization (WHO, Geneva, Switzerland) to evaluate the impact of “real-world” health services [11,12]. Hybrid type 1 studies test the effectiveness of clinical interventions while gathering information on its delivery in a real-world situation [12].

The study was drafted according to the Standards for Reporting Implementation Studies (StaRI) statement, a standardized reporting guideline containing 27 items [13].

### 2.2. Study Location

This study was conducted in the PHC of the Brazilian public health system (Unified Health System or *Sistema Único de Saúde* (SUS)) in the city of Belo Horizonte, which is the capital of the state of Minas Gerais (MG) and the sixth largest capital in Brazil with more than 2.5 million residents [14]. Belo Horizonte has 152 PHC centers and 80 pharmacists who provide clinical services to patients in the PHC centers.

PHC is the basis of the Brazilian universal public health system (SUS), provided through the same model throughout the country for Brazilians and foreigners on Brazilian soil in PHC centers. In the Brazilian public PHC, since birth, people are followed longitudinally throughout their lives by a family physician and by nurses. People also have access to consultations with multiple healthcare professionals from a multidisciplinary team, which may include a pharmacist, among other professionals (e.g., nutritionist, physical therapist, occupational therapist, physical educator) [15]. In the city of Belo Horizonte, pharmacists have been members of all PHC multidisciplinary teams since 2008, and provide consultations regarding medication use for patients. Pharmacists work closely with other healthcare professionals in PHC centers and may discuss the needs of their patients or refer their patients to them when deemed necessary. They are also authorized to prescribe over-the-counter (OTC) medicines.

### 2.3. Pharmaceutical Services for Smoking Cessation

Focusing on tobacco users, the pharmaceutical clinical activities in the Belo Horizonte PHC follow a Guideline published in 2018. Pharmacists are the only providers of smoking cessation services in the city of Belo Horizonte, however, as for any other patients, they can discuss tobacco users’ cases with other healthcare professionals or refer patients to them if necessary. 

For smoking cessation, the pharmacist can implement an intensive approach, in which an assessment of the motivational state and the dependence level of the patient is carried out, as well as the assessment of their individual needs, pharmacological and non-pharmacological, related to smoking. After the individual assessment, the patient undergoes periodic sessions of cognitive-behavioral therapy and follow-ups that can be carried out in an individual format (smoking cessation service with an individual approach); or in a group (smoking cessation service with a group approach). Generally, patients who are inserted in the individual approach are only those who could not be included in the group approach. The main reasons that justify the non-participation of a patient in the group approach are: problems with the group schedule; and health conditions that demand individualized care (e.g., pregnancy, uncontrolled mental illness) [16,17].

Up to ten sessions are held per approach, the first four being based on material provided by the Brazilian National Cancer Institute (*Instituto Nacional do Câncer* (INCA)), Rio de Janeiro, Brazil. The other sessions consist of discussions about various topics, including the participation of other health professionals [16,17]. In the services, smoking cessation is determined when the patient reports cessation to the pharmacists and does not present any abstinence symptoms (self-reported smoking cessation).

In addition, the following smoking cessation pharmacotherapy is distributed free of charge by the SUS: (1) nicotine transdermal patches (7, 14 and 21 mg) or chewing gum (2 mg), which can be prescribed by physicians, nurses or pharmacists (prescriptions provided by pharmacists began in June 2019) when no contraindications have been identified during the individual assessment performed by the pharmacist; (2) antidepressants, which are bupropion (150 mg tablet) or nortriptyline (25 mg capsule), prescribed exclusively by physicians during their regular PHC consultations if they deem it necessary. However, the use of nicotine therapy alone is preferred in smoking cessation services. Antidepressants are considered second-line treatment for most cases, unless patients have contraindications to nicotine use (e.g., pregnancy) or present other health conditions previously diagnosed (e.g., psychiatric disorders). The pharmacotherapy monitoring is performed by the pharmacist [16,17].

### 2.4. Data Source and Collection

All data were retrospectively collected and assessed for the period from January 2018 to December 2019. The data sources used were: GERAF (Pharmaceutical Assistance Management System), which is a software developed for the SUS of Belo Horizonte to manage the pharmaceutical services provided; and SISREDE, a stock management system, from which medication dispensing data were extracted.

### 2.5. Assessment of the Implementation of Pharmaceutical Services for Smoking Cessation

To assess the implementation of pharmaceutical services for smoking cessation, a descriptive analysis of its indicators was performed. The total number, mean and standard deviation were presented for the following variables: number of pharmacists offering the services; numbers of PHC centers where the services were offered; number of individual and group smoking cessation approaches; number of sessions carried out in individual and group approaches.

The following data related to the smoking cessation services were also described: approaches with the use of nicotine; approaches with the use of bupropion or nortriptyline (antidepressants); approaches that involved the prescription of medications by pharmacists; and approaches with at least four sessions.

### 2.6. Assessment of Pharmaceutical Services for Smoking Cessation Effectiveness

To assess the effectiveness of pharmaceutical services on smoking cessation (event of interest), the cessation reported by the patient (self-reported smoking cessation) to the pharmacist during an individual meeting or in the support group was considered. Cessation was described according to its incidence in the individual and group approach; and according to the number of sessions carried out until cessation.

The association between explanatory variables and the outcome event of interest (smoking cessation) was also assessed through univariate and multivariate analyses. The analyses were conducted by Poisson regression, which provides relative risk (RR) estimates with 95% confidence intervals (95%CI). Variables that presented *p*-value < 0.20 in the univariate analyses were added in the multivariate analysis. Multivariate modeling was also performed using Poisson regression, with those variables with a *p*-value < 0.05 remaining in the final model, a statistical criterion adopted to determine an independent and significant association between the explanatory variables and the event. The following explanatory variables were used: sex, use of pharmacotherapy for smoking cessation (no medication; *or* nicotine only; *or* one antidepressant only; or nicotine plus one antidepressant), number of sessions with the pharmacists (variable inserted in the analyses in its quantitative format), and dependence level.

The dependence level was measured according to the Fagerström Test for Nicotine Dependence. This test is used worldwide, and was adapted and validated in Brazil in 2002, with scores ranging from 0 to 10 [18]. For the purposes of univariate and multivariate analyses, the variable was divided into three categories after analyzing its distribution: 0–4—very low to low dependence level; 5–7—medium to high; and 8–10—very high.

All explanatory variables were also subject to descriptive analysis, with the determination of absolute and relative frequencies of qualitative variables and mean and standard deviation of quantitative variables. For all analyses, the Stata^®^ statistical package, version 12, StataCorp, College Station, TX, USA, was used.

## 3. Results

During the study period, sessions on smoking cessation were held in 93 of the 152 PHC centers (61.2%) by 65 of the 80 pharmacists (81.3%).

A total of 1761 smoking cessation approaches were performed: 170 individual (9.7%) and 1591 in group (90.3%). Among the approaches in the individual format, 32 consisted of at least four sessions (18.8%), with a total of 391 sessions performed (mean 2.3 ± 1.4 sessions per approach). Smoking cessation occurred in 39.4% of individual approaches (*n* = 67) (Figure 1).

Among the 1591 group approaches, 939 consisted of at least four sessions (59.0%), and of these, 58 reached ten sessions (3.6%). In total, 7075 sessions were carried out in the group approach (mean 4.4 ± 2.5 sessions per approach). For 712 of the approaches, smoking cessation was observed (44.8%) (Figure 2).

Most approaches were performed with female patients (*n* = 1150; 65.3%) and an average dependence level equivalent to 6.2 ± 2.0 was identified. Smoking cessation pharmacotherapy was used in most approaches (*n* = 1584; 89.9%): 748 (47.2%) involved the simultaneous use of nicotine and one antidepressant; 161 (10.2%) involved only the use of one antidepressant; and 675 (42.6%) the use of nicotine only. In 106 of the total 1423 approaches that involved the use of nicotine (7.4%), this medication was prescribed by the pharmacist (Table 1).

In the multivariate analysis, the use of nicotine plus an antidepressant (RR = 1.30; 95%CI = 1.08–1.57; *p* = 0.006) and the number of sessions with pharmacists (RR = 1.21; 95%CI = 1.19–1.23; *p* < 0.001) were independently and positively associated with smoking cessation, increasing the chance of cessation by 30% and 21%, respectively. On the other hand, the very high dependence level was negatively associated with smoking cessation (RR = 0.77; 95%CI = 0.67–0.89; *p* < 0.001), decreasing the chance of cessation in 23%.

## 4. Discussion

From the descriptive analysis of the indicators, one can note that the smoking cessation services were well implemented, being offered in most PHC centers (61.2%) and by most pharmacists (81.3%). However, there is potential for expansion in order to reach all PHC centers. The expansion and optimization of these services is relevant to the community, since this is a practice that has a positive impact, not only for the patient, but also for those who live with them. Furthermore, offering smoking cessation services in all PHC centers would probably mean being closer to the patients’ homes, facilitating their adherence to the services [19].

It is also important to increase the number of pharmacists providing the services in order to expand them by encouraging the other 18.7% of pharmacists who already work in PHC to initiate their activities in smoking cessation. For this, the training of current pharmacists and newly incorporated professionals must be guaranteed. The last training session on smoking cessation held in the SUS of Belo Horizonte was held in 2019. This may have limited the number of professionals offering their services over the studied period and reinforces the need for continuing education. The promotion, participation and support of permanent education actions are considered as a minimum standard for ensuring quality in pharmaceutical services and smoking cessation services in general [20,21].

A greater number of group approaches was observed as well as a higher proportion of smoking cessation with this type of approach (44.8% versus 39.4% in the individual approach). These results reinforce the idea that the group approach should be prioritized, as carried out in the assessed context. In addition, previous studies have shown that the group approach has better potential to lead to cessation, as it allows for the coexistence of tobacco users with similar experiences, which has a positive impact on the experience with the process [22,23,24].

This study also shows the increase in the proportion of smoking cessation with the rise in the number of sessions with pharmacists, both in the individual and group approach. This was also demonstrated from a global point of view, since, after approaches involving up to four sessions (*n* = 1065), the proportion of smoking cessation was 29.0%; in approaches involving more than four sessions (*n* = 696), however, the proportion of smoking cessation was 67.5% (results not shown). This demonstrates the importance of smoking cessation services and their successful performance by the pharmacist. The results also point to better effectiveness by performing a greater number of sessions than those recommended by the Health Ministry and INCA [25]. Such a longer format also allows for the motivational approach in the maintenance stage, which is essential to reduce relapse after initial cessation [26].

This notion was reinforced in the multivariate analysis, through which it was observed that, for each pharmacist session provided during the approaches, the incidence of smoking cessation increased by more than 20%. This result demonstrates that the pharmaceutical services offered in PHC have a high potential to impact the smoking cessation process, with the pharmacist being a key professional to directly contribute to the expansion and qualification of the National Tobacco Control Program. In European longitudinal studies, the effectiveness of pharmaceutical interventions in smoking cessation in private community pharmacies was also observed, but Brazilian studies with such an approach were not identified [27,28].

The use of nicotine associated with antidepressants also increased the incidence of smoking cessation according to the multivariate model (30% increase). Considering the multidimensionality of nicotine addiction, this result highlights the need for multiprofessional care, since the pharmacist is the professional responsible for monitoring the patient and prescribing nicotine; however, for the use of antidepressants, a physician’s prescription is necessary [29].

On the other hand, as expected, the very high dependence level (Fagerström score of 8–10) reduced the incidence of smoking cessation. In view of this association, seeking to improve and plan the services offered, the intensified intervention of the pharmacist is recommended in these cases, with the purpose of increasing the incidence of smoking cessation among these patients. Another option would be to further involve physicians and psychologists in the individual smoking cessation sessions of this particular group of patients, contemplating that partnership in new versions of the city’s smoking cessation guidelines.

A limitation of the present study is that the incidence of smoking cessation may have been overestimated in this study, since abstention was not determined by blood biochemical markers or by carbon monoxide breath monitor, but was only based on patient self-reporting. However, since this is a “real-world” study and service, it is important to point out that limited laboratory resources are available in the Brazilian public health system and testing for smoking cessation is not considered a priority in PHC. Additionally, the monitoring of smoking cessation using self-reported cessation complies with Brazilian guidelines [19]. Another limitation is the fact that cessation after 12 months from the initial cessation, which would indicate persistence of cessation, was not assessed. Thus, an active surveillance for patients with a high dependence level is recommended to assess the maintenance of cessation after the end of the approaches, making it possible to actively reinvite tobacco users on relapse to the services.

Another limitation of the present study stems from the quality of data sources. As these are secondary sources, some descriptive data that could have better explained the multivariate model were not available (e.g., age of patients and number of medications used). For the same reason, collecting data from a comparison group for the evaluated intervention was also not possible.

However, such limitations are counterbalanced by the fact that this study evaluates the results of “real-world” services, with a hybrid design, which is a format encouraged for assessing the impact of the implementation of services by WHO [11]. It is also important to point out that the present study describes the results of a smoking cessation program in a Brazilian PHC carried out by pharmacists, which, to our knowledge, is the first of its kind.

## 5. Conclusions

This study showed that the smoking cessation program offered by the SUS of Belo Horizonte contributes to reducing the use of tobacco in the studied population. The increase in the number of sessions provided by pharmacists and the combined use of nicotine with antidepressants was also verified to positively help in smoking cessation; while the high dependence level on nicotine impairs cessation.

Thus, the obtained results demonstrate that the promotion of qualified cessation with the pharmacist as the promoting agent was effective in the studied scenario. Considering the individual and collective benefits arising from the reduction in the prevalence of smoking, this study reinforces the continuing need for the expansion and qualification of pharmaceutical services, which, despite being effectively implemented, still have room for expansion in the PHC of Belo Horizonte.

## Figures and Tables

**Figure 1 ijerph-19-12305-f001:**
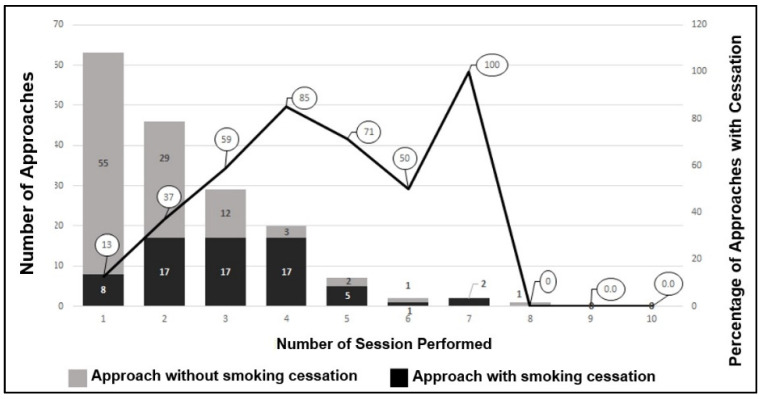
Incidence of smoking cessation in the pharmaceutical service with individual approaches. 2018–2019. Belo Horizonte—MG. Brazil.

**Figure 2 ijerph-19-12305-f002:**
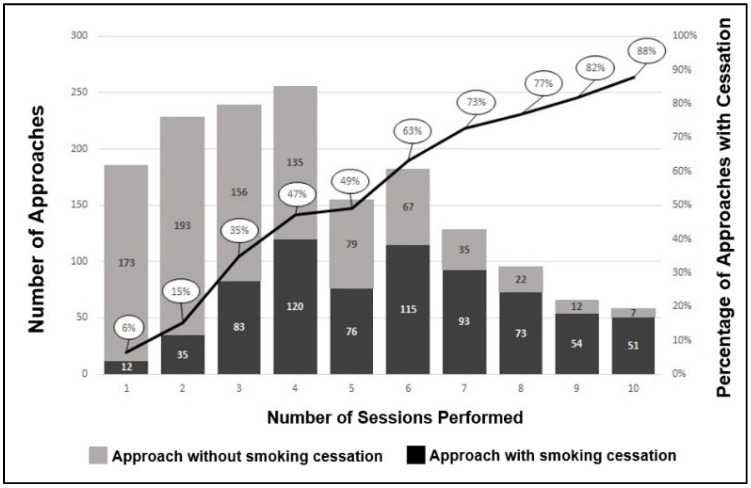
Incidence of smoking cessation in the pharmaceutical service with group approaches. 2018–2019. Belo Horizonte—MG. Brazil.

**Table 1 ijerph-19-12305-t001:** Factors associated with smoking cessation according to univariate and multivariate analyses. 2018–2019. Belo Horizonte—MG. Brazil.

Variable	Cessation		Univariate Analysis		Multivariate Analysis	
	Yes	No	RR (95%CI) #	*p*-Value ##	RR (95%CI) #	*p*-Value ##
**Sex—N (%)**						
Female	508 (44.2)	642 (55.8)	1	-	-	-
Male	271 (44.4)	340 (55.6)	1.00 (0.90–1.12)	0.942	-	-
**Medication use—N (%)**						
None	59 (33.3)	118 (66.7)	1	-	1	-
Nicotine only	273 (40.4)	402 (59.6)	1.21 (0.97–1.52)	0.096	1.09 (0.90–1.33)	0.364
Antidepressant * only	60 (37.3)	101 (62.7)	1.12 (0.84–1.49)	0.450	1.09 (0.85–1.40)	0.477
Nicotine and antidepressant *	387 (51.7)	361 (48.3)	1.55 (1.24–1.93)	<0.001	1.30 (1.08–1.57)	0.006
**Dependence level**						
0–4—very low to low	146 (46.6)	167 (53.4)	1	-	1	-
5–7—medium to high	446 (45.5)	534 (54.5)	0.98 (0.85–1.11)	0.724	0.90 (0.80–1.02)	0.110
8–10—very high	187 (40.0)	281 (60.0)	0.86 (0.73–1.01)	0.062	0.77 (0.67–0.89)	0.001
**Number of sessions with pharmacists ****						
	5.5 (2.4)	3.2 (2.0)	1.21 (1.19–1.23)	<0.001	1.21 (1.19–1.23)	<0.001
**Mean (standard deviation)**						

* Antidepressant = use of bupropion or nortriptyline; ** Variable analyzed in quantitative format; # RR (95%CI) = relative risk and 95% confidence interval; ## *p*-value = based on Poisson regression.

## Data Availability

Restrictions apply to the availability of these data. Data were obtained from the SUS of Belo Horizonte and are available upon request from the Research Nucleus of the City Government of Belo Horizonte.

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
