# Peer review of "Assessment of Pharmaceutical Services for Smoking Cessation: An Effectiveness–Implementation Hybrid Study"

_ijerph, 2022, doi:10.3390/ijerph191912305_

Round 1

Reviewer 1 Report

Dear authors,

thank you for this study, highlighting the potential relevance of the pharmacist in delivering smoking cessation treatment in primary care services in Brazil.

There are some points to be clarified.

In the abstract you talk about pharmaceutical services in primary health care (PHC). In Brazil pharmacists are professionals working in PHC and prescribing and administering medicines, like doctors?

Lines 44-45: specify that those numbers are related to the world.

Line 54: Could you specify/define what a pharmacist in Brazil can do and its role, for example in other countries pharmacists are working in drug stores and can only administer counselling/brief advice and give over the counter nicotine therapy but not other medicines that need to be prescribed by a doctor.

Line 63: correct “The was drafted”

Lines 80-92: the pharmacist does all the activities described in this part? No doctor or psychologist are involved? You should describe better how this service is organized and when or in which cases the other professionals contribute to the pharmacist activity or when their intervention is requested. What is the level of their collaboration?

Lines 186-189: in this description it appears that not only the pharmacist but also a doctor prescribed medicines in particular antidepressants (which cannot be prescribed by the pharmacist). How it happens that a pharmacist prescribes nicotine without the support of a doctor? In the cases that are easier to manage with only nicotine? When is it requested the medical intervention, when nicotine alone is not effective for cessation? When is it the case that a doctor is needed to prescribe only antidepressant? Is it a pharmacist decision to involve the doctor when nicotine is not effective? You should explain more the procedure followed by the professionals and how they collaborate in the decisions to be taken for the treatment.

Line 195: explain why a very high level of dependence was negatively associated with smoking cessation.

Lines 207-208: Why is it so important to expand the number of pharmacist, compared to the doctors or psychologists?

Lines 250-252: why do you give so much relevance to the pharmacist in treating the patients with high level of dependence. On the contrary, for these cases it should be important the collaborative intervention of all the professionals including doctors and psychologists. Explain why do you think that a pharmacist would be able to successfully treat these patients compared to the collaboration with other professionals, considering also the limitations of this study that include that smoking abstinence was not ascertained after 12 months and the self-reporting.

Author Response

We thank the reviewer for all comments and suggestions.

Please see atachment.

Reviewer 2 Report

The paper demonstrates that pharmacists working with primary health care can deliver smoking cessation medications and counselling effectively.  The reported cessations rates are higher than usually reported and the authors do not define cessation in the paper.  They acknowledge that cessation is not confirmed biochemically which is a weakness of the paper.  No definition is provided of "cessation".  When is it determined?  How long does the individual need to be abstinent? Who/how is abstinence verified

They describe their work as a type 1 effectiveness implementation hybrid.  It would helpful to many readers to describe what is meant by this type of study.  This could be addressed in the last section before the Conclusions

Relapses are common as tobacco use is an addiction. The authors should provide information on relapse within their study and discuss the issue of relapse in the discussion.

Additional minor points:

The term "smoker" is considered by many to stigmatize individuals wo smoke.  One suggestion is to refer to "smokers" as users of cigarettes or tobacco.

line 63 a word is missing after "The"

In section 2.2, second paragraph, would suggest "a" pharmacist rather than "the"

line 209 should read "session" not "section" 

Finally the use of the word "habit" (Line 273) is inappropriate as tobacco use is an addiction

Author Response

We thank the reviewer for all comments and suggestions.
